# Predictions for Three-Month Postoperative Vocal Recovery after Thyroid Surgery from Spectrograms with Deep Neural Network

**DOI:** 10.3390/s22176387

**Published:** 2022-08-24

**Authors:** Jeong Hoon Lee, Chang Yoon Lee, Jin Seop Eom, Mingun Pak, Hee Seok Jeong, Hee Young Son

**Affiliations:** 1Division of Biomedical Informatics, Seoul National University Biomedical Informatics (SNUBI), Seoul National University College of Medicine, Seoul 110799, Korea; 2Department of Otolaryngology, Thyroid/Head & Neck Cancer Center, The Dongnam Institute of Radiological & Medical Sciences (DIRAMS), Busan 46033, Korea; 3Samsung Electronics Co., Ltd., 130 Samsung-ro, Yeongtong-gu, Suwon-si 16677, Korea; 4Microsoft, Redmond, WA 98052, USA; 5Department of Radiology, Pusan National University Yangsan Hospital, Yangsan 50612, Korea

**Keywords:** deep learning, voice recovery, spectrogram, GRBAS

## Abstract

Despite the lack of findings in laryngeal endoscopy, it is common for patients to undergo vocal problems after thyroid surgery. This study aimed to predict the recovery of the patient’s voice after 3 months from preoperative and postoperative voice spectrograms. We retrospectively collected voice and the GRBAS score from 114 patients undergoing surgery with thyroid cancer. The data for each patient were taken from three points in time: preoperative, and 2 weeks and 3 months postoperative. Using the pretrained model to predict GRBAS as the backbone, the preoperative and 2-weeks-postoperative voice spectrogram were trained for the EfficientNet architecture deep-learning model with long short-term memory (LSTM) to predict the voice at 3 months postoperation. The correlation analysis of the predicted results for the grade, breathiness, and asthenia scores were 0.741, 0.766, and 0.433, respectively. Based on the scaled prediction results, the area under the receiver operating characteristic curve for the binarized grade, breathiness, and asthenia were 0.894, 0.918, and 0.735, respectively. In the follow-up test results for 12 patients after 6 months, the average of the AUC values for the five scores was 0.822. This study showed the feasibility of predicting vocal recovery after 3 months using the spectrogram. We expect this model could be used to relieve patients’ psychological anxiety and encourage them to actively participate in speech rehabilitation.

## 1. Introduction

Voice disorders, the most common discomfort, are known to occur in 40% to 80% of patients who undergo thyroid surgery [1,2]. Vocal cord paralysis, which is caused by nerve damage, including damage to the recurrent laryngeal nerve, receives the most medical attention. However, it is very rare and found only in about 1% of patients, while functional vocal problems that occur without special anatomical abnormalities after thyroid surgery have been reported to occur about 30% of the time [1,2]. After a thyroidectomy, although patients may have normal findings in terms of their laryngeal endoscopy, it is common that they may complain of vocal dysfunction [3,4]. In many cases, it is difficult to find the cause of the change in speech by way of visual observations such as performing a laryngoscope examination.

The typical symptoms of patients who complain of vocal dysfunction after thyroid surgery are vocal fatigue, difficulty producing treble, difficulty maintaining vocalization, difficulty breathing, mild hoarseness, and problems with swallowing [5]. Functional voice disorders are characterized by dysphonia in the presence of apparently normal vocal fold anatomy and movement. Choi et al., reported that even in the absence of laryngeal nerve damage, half of the patients may experience vocal changes, which require a 6-month recovery period [1]. The ability to identify early changes would facilitate an early referral for a comprehensive voice evaluation that aims at improving the patient’s quality of life, preventing secondary injuries, and identifying those who might benefit from vocal fold augmentation [6].

The GRBAS scale has been established as the most compact perceptual grading system [7]. The perceptual GRBAS scale is reliable, and has been applied worldwide. Its use has enabled a better level of communication among clinicians and speech-language pathologists concerning their patients’ voices, which constitutes a major advantage for an assessment tool [8].

In otorhinolaryngology, because the vocal samples are collected using the same protocol, quantitative analysis is possible in terms of using objective and quantitative data [9,10]. The acoustic sound generated through acoustic-voice inspection can be used as various types of data by preprocessing it. Studies reported that the normalization of speech data into spectrograms can be used for classification and regression [11]. There have been several studies using a spectrogram that transforms voice data into an image form using a convolutional neural network (CNN) algorithm [12,13,14]. Moreover, the GRBAS score, a subjective measure among the vocal analysis methods, was predicted by researchers using machine learning techniques [15].

With the advent of machine learning technology, the use of artificial intelligence (AI) in medicine has become essential [16]. Particularly in thyroid cancer, deep learning has been widely used for the diagnosis of disease and the detection of lesions [17,18,19]. Because vocal diseases have a great impact on a patient’s quality of life, predicting the prognosis is crucial [20,21]. However, despite advances in acoustic-acoustic testing, AI studies using audio data are relatively rare in medicine.

This study aimed to predict vocal recovery after 3 months by using a deep neural network algorithm from preoperative and postoperative vocal spectrograms. This can improve the patient’s quality of life and allow them to receive treatments in advance. The performance of the model was verified through internal consecutive split validation, and the possibility of an application to AI’s negative disease has been examined.

## 2. Related Works

Our study has several differences compared with previous related studies on postoperative outcomes [22,23,24]. There have been studies to find prognostic factors related to the voice outcome. However, these were studies analyzing the prognostic factors related to voice, not AI-based prognosis prediction models. In this study, the goal was to predict long-term outcomes in advance using the patient’s pre- and postoperative voices with deep learning. Furthermore, previous studies on using acoustic samples for voice disorder prediction have focused on the diagnosis of the current condition [25,26,27]. By contrast, in our study, we predicted the patient’s prognosis and quickly suggested treatment to patients with poor prognoses. Providing these clinical decisions will help improve patients’ quality of life.

## 3. Materials and Methods

### 3.1. Patients and Vocal Data

The protocol for this retrospective study was approved by the Ethics Committee of the Institutional Review Board (D-1801-003-002) at Dongnam Institute of Radiological & Medical Sciences (DIRAMS). Written or oral informed consent was not obtained from the patients because this study had a nonintrusive retrospective design and the IRB waived the need for individual informed consent because all data were anonymously analyzed. All experiments were performed in accordance with relevant guidelines and regulations, and all experimental protocols were approved by DIRAMS. The vocal data analyzed during the current study are not publicly available, but are available from the corresponding author upon reasonable request.

Our exclusion criteria were as follows: first, the patients with previous laryngeal or vocal fold paralysis; second, patients with pulmonary diseases; third, patients who had previous neck surgery; fourth, patients who showed post-thyroidectomy evidence of recurrent laryngeal nerve; and, finally, patients with an injury of the external branch of the superior laryngeal nerve palsy. Patients with preoperative, postoperative, and after-3-months voice samples were recruited in this study.

We consecutively enrolled 114 patients with a thyroidectomy who were diagnosed with thyroid cancer and who had visited the hospital from January 2018 to December 2019. The average age was 47.4 years, and the sex split was 26 men and 88 women. The distribution of grade scores for all patients can be found in Table 1.

To measure the performance of the model, internal consecutive split validation was performed, and the test set was divided by time. Of the total data, 92 sample data, corresponding to 80%, were used for data training and for the parameter tuning set, and 22 sample data recruited last, corresponding to 20%, were used as a test set.

To evaluate the degree of patients’ speech impairment, the GRBAS scale, a perceptual evaluation method, was applied to compare the degree of speech impairment in each session [28]. The GRBAS system for describing vocal quality contains five well-defined parameters: G (overall grade of hoarseness), R (roughness), B (breathiness), A (asthenic), and S (the strained) quality of the voice [7].

### 3.2. Preprocessing the Vocal Data

The speech signals in the vocal data—preoperative, postoperative, and 3 months postoperative—were all generated through the same protocol. The speech signal in the voice data with durations ranging from less than 1 s to about 20 s was sampled at 11,025 Hz. Each sentence was labeled with the GRBAS score by an experienced phonologist. We calculated the spectrograms for all vocal data. We ignored the rest and used a frequency range of 0–6 KHz. The DFT was calculated through a frame size of 1024. The DFT data were converted to the log-power spectrum. Finally, through RGBA mapping, spectrogram images of the same size were created. The model architecture and the process of generating the spectrogram are included in the Appendix A.

### 3.3. The Two Stages of Deep Neural Networks

To predict the amount of vocal recovery after 3 months using pre- and postoperative vocal data, we first developed a pretrained model to predict the GRBAS score. To evaluate the performance of predicting the GRBAS score from the voice sample, the performance was measured using a 20% random split. The architecture of the deep neural network was based on EfficientNet-B4, which is a recent state of the art structure [29]. We added two fully connected layers to the last layer of the CNN model, composed of 1024 and 5 units for the final layer, respectively. The mean squared error (MSE) was used as the loss function, and the model was compiled to predict the GRBAS score as a label for the final layer with five units. The model was used with the parameters pretrained from ImageNet data, and the fine-tuning of the entire model was performed using all data.

The above model was used as an encoding model to extract the important features to predict the GRBAS score as a pretrained model. Two CNN models were connected by separately applying models that extracted 1024 features to pre- and postoperative vocal data. We concatenated two feature vectors to compose the input of the next long short-term memory (LSTM) layer, which is a variant of the recurrent neural network involving time series by introducing a three gates operation (input, output, and a forget gates) [30]. Then, we used the LSTM recurrent neural network for the time-series prediction at the postoperative and 3-month times from pre- and postoperative spectrograms [31]. Finally, this model architecture was fine-tuned to predict the postoperative GRBAS scores and GRBAS score after 3 months without a spectrogram after 3 months with an MSE loss of 25. The scheme of the overall model structure is shown in Figure 1, and the detailed structure and source code of the model are shown in Appendix A.

To avoid the overfitting, online augmentation was performed by transforming the images while learning. During the learning process, the rotation range within 20 degrees, the 20% range for the width and height, and the shear zoom were randomly applied to the training images. This process was performed using ImageDataGenerator supported by Tensorflow Keras (Google, Mountain View, CA, USA). However, the flip, which could change the meaning of the spectrogram, was not used.

We used the Keras deep-learning framework included in Tensorflow version 2.3.0 (Google, Mountain View, CA, USA). We trained the model using the Adam optimizer to accelerate the convergence of network parameters with a learning rate of 0.001. The learning rate was gradually reduced through the call-back function. The batch size was 24. A total of 200 epochs were performed, and the weight with the lowest validation loss was used. We used EfficientNet-B4 as the backbone architecture of the CNN (https://github.com/qubvel/efficientnet—accessed on 16 January 2022).

### 3.4. The Activation Heatmap of the Convolution Layer

We inferred the activation heatmap of the model that predicted the GRBAS score to identify the important location in the spectrograms. To derive the activation heatmap, we used the gradient-weighted class activation mapping (Grad-CAM) method using gradient-based localization [32]. The heatmap was derived by computing the gradients for class with respect to the feature map of the selected convolutional layer. The activation heatmaps were derived from the convolutional layer in the second and third residual block, corresponding to 56 × 56 and 28 × 28 of the model, respectively.

### 3.5. Performance Evaluation

To measure the performance of the model, internal consecutive split validation was performed with the test set divided by time. The final output was the GRBAS score value, the MSE values were calculated for each label in 22 patients, and the Spearman’s rank correlation test was performed. To measure the area under the receiver operating characteristic curve (AUROC), the GRBAS scores were divided into binary values of 0 or greater than 1.

In Equation (1), *n* and *i* denote the total number of patients and the index of the patient’s vocal sample, respectively. *t* and *s* denote the time point of the vocal samples (preoperative, postoperative, and 3-months-after vocal sample). This metric was calculated in terms of the five scores representing the GRBAS. The Spearman’s rank correlation coefficient, used for the performance evaluation of numeric labels, was computed for the derived parameters according to Equation (2), where *n* and *d* denote the number of voice samples and the rank difference between the GRBAS score and predicted GRBAS score, respectively.
(1)$$RMSE_{ts}=\sqrt{\frac{1}{n}\overset{n}{\underset{i=1}{\sum }}{\left(y_{its}-{\tilde{y}}_{its}\right)}^{2}}$$
(2)$$\rho =1-\frac{6\sum d_{i}^{2}}{n\left(n^{2}-1\right)}$$

## 4. Results and Discussion

### 4.1. The Prediction Performance of the Deep Neural Network

Table 2 summarizes the performance of the deep neural network predicting GRBAS after 3 months of surgical treatment using pre- and postoperative voice spectrograms. The overall root mean square error (RMSE) value of the GRBAS score of the deep neural network was 0.3798. The RMSE values for each GRBAS score was 0.399, 0.365, 0.409, 0.469, and 0.203. In the Spearman’s rank correlation test, the grade was 0.741, roughness was 0.153, breathiness was 0.766, and asthenia was 0.433; the strain could not be calculated because all test samples had zeros. Only two patients had a value of one for roughness, and all others were zero. Where the Spearman’s rank correlation results were concerned, scores that had a statistically significant correlation were grade, breathiness, and asthenia (*p* < 0.01). For the roughness of the test sample, only two samples had a score of one, and the rest were zero. Figure 2 shows the relationship between the prediction results for grade, breathiness, and asthenia for the test set and the actual score.

We used EfficientNet-B4 among the CNN architectures. The changes in performance when using other CNN models are summarized in Appendix A. The efficientnet-b4 architecture had the lowest RMSE value in terms of the average value of all scores.

### 4.2. Binary Classification Performance for Prognostic Prediction

Because RMSE and correlation have different meanings that depend on the range and scale of the data, they do not represent the performance of an objective model. Therefore, we divided the grade, breathiness, and asthenia scores into binary values of zero or greater than one, and measured AUROC scores using these (Figure 3). The AUROC values were 0.894, 0.918, and 0.735 for grade, breathiness, and asthenia, respectively. The roughness score was more than one among 22 patients used as the test set, and the AUROC value was 0.575, which was poorly predicted. The strain score was zero for all patients in the test set. Because the strain score was zero for all patients in the test set, it was impossible to calculate the AUROC value.

The optimal thresholds corresponding to the Youden index yielded 76.5% and 100.0% sensitivity and specificity for the grade score. For breathiness and asthenia, sensitivity and specificity were 70.6% and 100.0%, and 55.6% and 92.3%, respectively.

### 4.3. Prognosis in Patients after 6 Months

Among the consecutively recruited patients, vocal samples after 6 months were obtained for 12 of 22 patients in the test set. The distribution of GRBAS scores of patients 6 months after surgery is shown in Appendix A. The AUROC values were 0.852, 0.800, 0.688, 0.861, and 0.909 for grade, roughness, breathiness, asthenia, and strain, respectively (Figure 4).

The optimal thresholds given by the Youden index, sensitivity and specificity of the grade score, were 78.8% and 100.0%, respectively. For roughness and breathiness, sensitivity and specificity were 60.0% and 100.0%, and 62.5% and 75.0%, respectively. For asthenia and strain, sensitivity and specificity were 100.0% and 83.3%, and 100.0% and 90.9%, respectively.

### 4.4. Scores for the Activation Heatmap

To visualize the important features of the scores in the spectrograms, an activation heatmap was constructed using the second and third residual block of the EfficientNet model (Figure 5). The voice spectrogram of a patient with high-degree grade was shown in Figure 5A. This spectrogram is from one of the patients with very poor voice quality. In the heatmap visualization through Grad-CAM, the highlighted part shows an important imaging feature to predict the grade score. The spectrogram of a normal grade voice is shown in Figure 5B. In this patient, the amplitude widely spreads over various frequencies. The region where the heatmap is activated in the spectrogram is very narrow.

### 4.5. Discussion

This study aimed to predict patients who will have problems with long-term vocal recovery. We obtained patient voice samples for those who underwent surgical treatment as well as their GRBAS scores from experienced phonologists for three points in time: preoperative, postoperative, and 3 months after surgery. Using this data, we developed a model that predicts, using deep learning, the GRBAS score of the patient’s voice 3 months after the operation. For model performance, the RMSE value was 0.3798, and the results for the Spearman’s rank correlation analysis for the grade, breathiness, and asthenia scores were 0.796, 0.784, and 0.602, respectively. When each score was binarized, the AUROC values for grade, breathiness, and asthenia were 0.894, 0.918, and 0.735, respectively. Through these results, we showed the possibility of predicting a patient’s long-term vocal disorder using the pre- and postoperative voice samples. This method is expected to help relieve patients’ psychological anxiety and encourage patients to actively participate in speech rehabilitation.

There have been studies conducted to predict voice quality and voice disorders using deep learning from voice samples [33,34,35]. The results of these studies have demonstrated the potential of using predictive models in clinical settings, which have shown high performance comparable to human evaluation. This study aimed to predict voice quality after 3 months from pre- and postoperative vocal samples. Long-term voice change after 3 months can be used for prognosis prediction, a problem that cannot be evaluated by humans. To the best of our knowledge, this is the first study predicting the prognosis of long-term voice quality after surgery.

Evaluating the objective patient’s voice is a challenging task because it is laborious, time-consuming, and cost-intensive [36]. Additionally, it is difficult to objectively measure voice samples given that interobserver variability may exist depending on the phonetician, and intraobserver variability may exist depending on the condition [7,37]. To provide an objective measure for this, there have been various methods that predict GRBAS by using machine learning. In our study, we implemented a deep-learning model that predicts the voice after 3 months postsurgery based on a pretrained model, which is the feature extractor based on the GRBAS score.

The results for predicting vocal diseases 3 months after a patient’s surgery can be regarded as a prognostic marker. Long-term vocal defects are a major factor in a patient’s quality of life, and it is almost impossible to predict them in advance. In this study, we found that by using internal consecutive split validation, AI models could predict long-term vocal defects. This means that the patient’s voice, as generated by the same protocol, can be used for objective data. Therefore, we suggest the possibility that a patient’s voice data can be applied to various purposes to make a prognosis for the patient.

Our study has some limitations. Our data ere from 112 people, which is not enough to train a deep neural network. Next, we are planning a validation study by recruiting more patients with long-term follow-up. We did not use factors that could have affected interpersonal phonetic outcomes, such as the patient’s surgical range, method, or age. Because the variability in the GRBAS score varies depending on the person who measures it, predicting it cannot completely cover the vocal condition. To find an objective answer to this problem, we first constructed a pretrained model that extracts features based on GRBAS, and we attempted to avoid the problem by applying a deep neural network to pre- and postoperative data.

In conclusion, we developed a model for predicting the recovery of a patient’s voice after 3 months by using a deep neural network algorithm from pre- and postoperative voice spectrograms. This approach can help physicians select patients with long-term vocal disorders for whom intensive care should be applied.

## Figures and Tables

**Figure 1 sensors-22-06387-f001:**
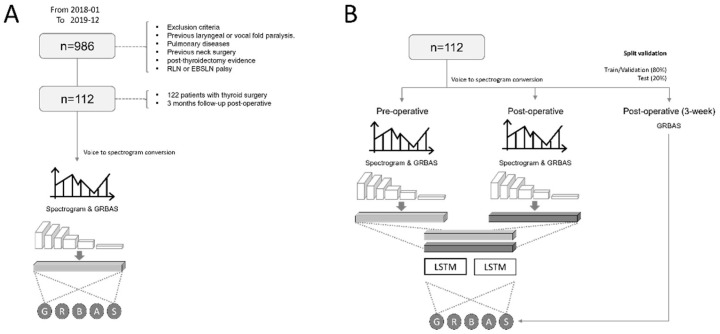
Workflow scheme of the data-learning process for acoustic vocal samples of patients with thyroid surgery using an artificial intelligence model. (**A**) Inclusion criteria and steps to train a pretrained model that extracts important features from vocal data. (**B**) The development of a deep learning model to predict the GRBAS score after three weeks from pre- and postoperative vocal samples with the pretrained model.

**Figure 2 sensors-22-06387-f002:**
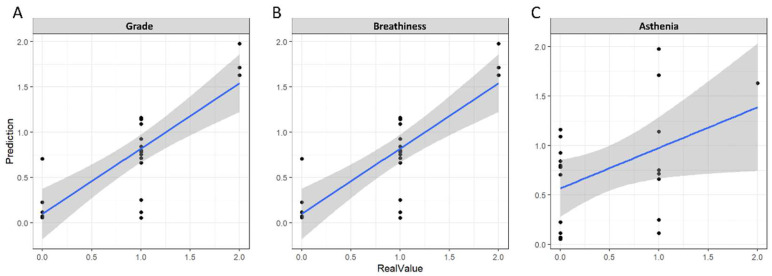
The prediction results for the (**A**) grade, (**B**) breathiness, and (**C**) asthenia scores of the test set. The x-axis represents the observed GRBAS score of the patient, and the y-axis represents the predicted value by the deep learning model. The blue line is the regression line to see the relationship between the predicted value and the actual value.

**Figure 3 sensors-22-06387-f003:**
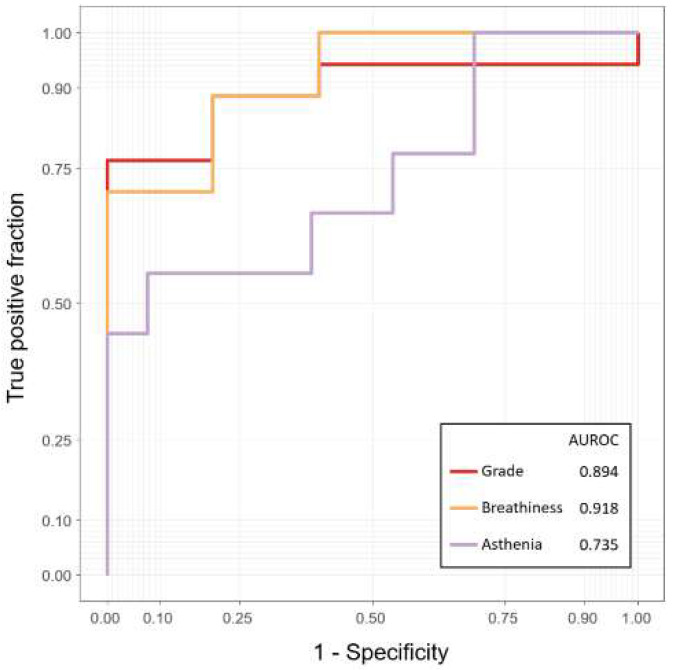
For the grade, breathiness, and asthenia scores, we divided the patient based on 0 or not, and the ROC was calculated.

**Figure 4 sensors-22-06387-f004:**
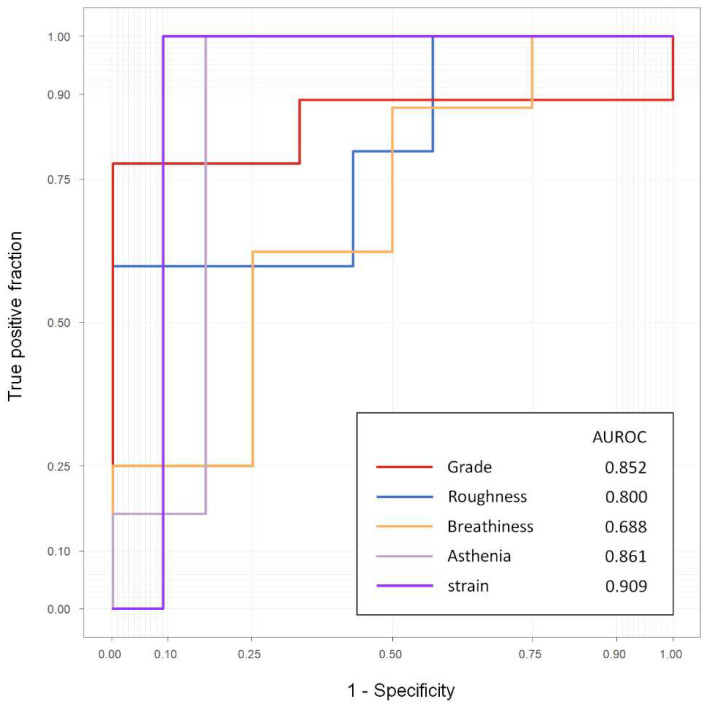
ROC curve for vocal samples of patients 6 months after surgery.

**Figure 5 sensors-22-06387-f005:**
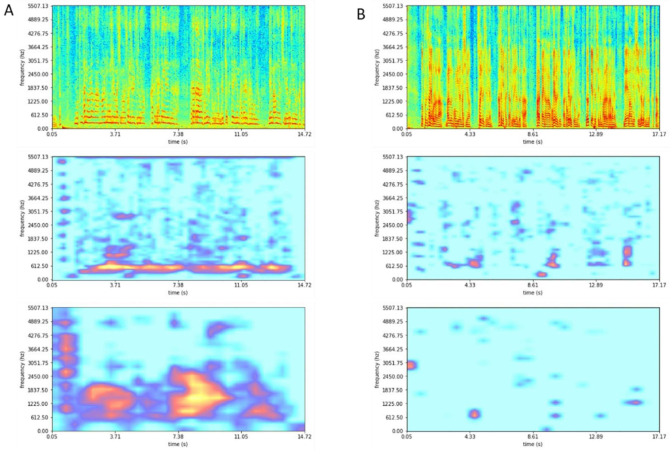
A spectrogram example and its visualization results using gradient-based localization to predict the grade score in second and third residual block of EfficientNet. (**A**) Spectrogram with heatmap visualization of a patient with high-degree grade score. (**B**) A patient with normal grade.

**Table 1 sensors-22-06387-t001:** Grade score distribution.

Grade	Pre op.	Post op.	3 Months Post op.
**G0**	43	25	31
**G1**	61	60	67
**G2**	9	24	14
**G3**	1	5	2

**Table 2 sensors-22-06387-t002:** Prediction performance for the GRBAS score.

Class	RMSE	Rho	*p* Value
Grade	0.399	0.796	<0.001
Roughness	0.365	0.149	0.509
Breathiness	0.409	0.784	<0.001
Asthenia	0.469	0.602	0.003
Strain	0.203	NA	NA

## Data Availability

The vocal data analyzed during the current study are not publicly available, but are available to the corresponding author upon reasonable request.

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
