# Peer review of "Predictions for Three-Month Postoperative Vocal Recovery after Thyroid Surgery from Spectrograms with Deep Neural Network"

_sensors, 2022, doi:10.3390/s22176387_

Round 1

Reviewer 1 Report

The paper presents application of neural networks to create model for prediction of the recovery of the patient’s voice after 3 months from preoperative and postoperative voice spectrograms.

1. In my opinion the paper is out of the scope of the Sensors Journal - it has nothing to do with sensors.

2. What is presented is not a research, but the case study of the use of well-known methods to create a model to predict some data. It is a basic engineering task.

3. Application of static neural networks for a dynamical process, which is the state of a recovering patient changing over time, is a wrong approach.

4. Too little amount of data is used for this application, both 92 for learning and  22 sets of data for tests is not sufficient to prove the performance of the model.

Finally, the paper is written in the disorganised manner, i.a. the supplementary materials mentioned in the text are not attached to the paper; the paragraph Discussion should discuss obtained results, but it contains introduction part, summing up part, references to the literature that should be cited in the introduction; some parts are unclear, some sentences are repeated a few times.

Author Response

The paper presents application of neural networks to create model for prediction of the recovery of the patient’s voice after 3 months from preoperative and postoperative voice spectrograms.

R1-1) Overall, thank you so much for the detailed review. We have carefully reviewed the errors you pointed out.

1. In my opinion the paper is out of the scope of the Sensors Journal - it has nothing to do with sensors.

R1-2) We submitted our article to "Biomedical Signal and Image Processing in Speech Analysis", one of the issues with sensors. Our study predicts the prognosis after surgery using voice signals, which is related to this issue.

2. What is presented is not a research, but the case study of the use of well-known methods to create a model to predict some data. It is a basic engineering task.

R1-3) As you said, deep learning technology itself is just an application. However, it is a novel study in the medical field, especially laryngoscope, to apply the deep learning model to predict prognosis after surgery through long-term follow-up of patients. We recruited 114 patients and presented the analysis results of a 6-month follow-up study.

3. Application of static neural networks for a dynamical process, which is the state of a recovering patient changing over time, is a wrong approach.

R1-4) Acquiring patient data in hospitals is very difficult. According to our hospital's protocol, the period of obtaining a patient's voice sample is fixed, so we used conventional CNN and LSTM methods. 

4. Too little amount of data is used for this application, both 92 for learning and 22 sets of data for tests is not sufficient to prove the performance of the model.

R1-5) It is not easy to obtain data from 6 months of follow-up of the patient's voice after surgery. In the two studies reviewed in the <related works> chapter, the number of patients was 39 for the article in [1] and 32 for the article in [2].

[1] https://doi.org/10.1007/s00405-016-4163-6 (n=39)

[2] https://doi.org/10.1002/lary.28282 (n=32) 

Finally, the paper is written in the disorganised manner, i.a. the supplementary materials mentioned in the text are not attached to the paper; the paragraph Discussion should discuss obtained results, but it contains introduction part, summing up part, references to the literature that should be cited in the introduction; some parts are unclear, some sentences are repeated a few times.

R1-6) Thank you for your detailed review. We attached supplementary methods and materials, and modified the structure of manuscript.

Reviewer 2 Report

Abstract

The title is showing Deep CNN model while in abstract written deep-learning model with a LSTM. Check properly.

Introduction

The literature review is very poor. The motivation is clear however, the problem statement and objective of the proposed study is not clear. I suggest highlighting the objectives of this study from the literature review more clearly.

Materials and Methods 

The choice of proposed methodology flowchart for more clear understanding.

A few stages in Figure 1 are not clear.

Correlation analysis of data need to be added here

Few parameters in Equations 1 and 2 are not defined

Results

In Table 1, check is it rho value or p value?

Resolution of Fig 5 need to be improved

Discussion

This section can be combined with Results section

The comparison of proposed method with various other techniques in literature is missing

Check Sections 5 and 6. Looks not proper.

Also check Supplementary Materials, Institutional Review Board Statement, Informed Consent Statement, Data Availability Statement, Acknowledgments, the authors haven't added anything simply used the content of MDPI template.

Appendix A and Appendix B also the same mistake.

Author Response

Abstract

The title is showing Deep CNN model while in abstract written deep-learning model with a LSTM. Check properly.

R2-1) Thank you for your comments. We modified the title. 

“Predictions for Three Months Postoperative Vocal Recovery after Thyroid Surgery from Spectrograms with Deep Neural Learning”

Introduction

The literature review is very poor.

The motivation is clear however, the problem statement and objective of the proposed study is not clear.

I suggest highlighting the objectives of this study from the literature review more clearly.

R2-2). Thank you. We modified the final paragraph of Introduction section, and added new paragraph <2. Related works> with additional references. 

Materials and Methods 

The choice of proposed methodology flowchart for more clear understanding.

A few stages in Figure 1 are not clear.

R2-3) We modified the figure 1.

Correlation analysis of data need to be added here

Few parameters in Equations 1 and 2 are not defined

R2-4) Yes, we did.

Results

In Table 1, check is it rho value or p value?

R2-5) Thank you for your detailed review. Yes, we modified the Cor to Rho.

Resolution of Fig 5 need to be improved

R2-6) Re-uploaded high-resolution photos.

Discussion

This section can be combined with Results section

R2-7) Yes, we did.

The comparison of proposed method with various other techniques in literature is missing

R2-8). Thank you. Literatures related to this study <2. Related works> with additional references. 

Check Sections 5 and 6. Looks not proper.

R2-9) Yes, we did.

Also check Supplementary Materials, Institutional Review Board Statement, Informed Consent Statement, Data Availability Statement, Acknowledgments, the authors haven't added anything simply used the content of MDPI template.

Appendix A and Appendix B also the same mistake.

R2-10) Yes, we did.

Reviewer 3 Report

Please find the reviewer's comments in the PDF file.

Round 2

Reviewer 1 Report

The paper was corrected, however in my opinion, what is presented is still not a research, but the case study of the use of well-known methods to create a model to predict some data. It is a basic engineering task - applying known methods to solve a problem. If "it is a novel study in the medical field" it should be reported in a medical journal, because it brings nothing to the deep learning algorithms issue. It is just another application of neural networks for image processing and feature extraction. It can also be presented on a conference.

The amount of data will not increase or improve because someone else has collected less data. These numbers are too low to statistically prove the performance of the model, especially when speech signal sample is concerned.

Author Response

R1-1) We submitted our article to the journal, Sensors with issue "Biomedical Signal and Image Processing in Speech Analysis". We tried to find appropriate journals about this study, and we found the Journal 'Sensors' has published a lot of biomedical researches.

R1-2) We did our best to ensure the number of patients who collected voice samples after thyroid cancer surgery. However, it was very difficult to recruit a sufficient number of patients and samples due to the nature of this medical field, and this was added to the limitation.

Discussion section (318 - 320 lines): Our study has some limitations. Our data is from 112 people, which is not enough to train a deep neural network. Next, we are planning a validation study by recruiting more patients with long-term follow-up.

Reviewer 2 Report

1. The title of the paper has been changed as per the suggestion. However, in the submission portal (SuSy), the title has not been changed. Also authors simply changed the title the contents of the paper has not been verified.

2. The parameters in Equations 1 and 2 are not defined

3. No details in section 3.4

4. Provide more details about LSTM

5. Define GRBAS

Reviewer 3 Report

All comments from the reviewer have been solved. The paper can be accepted in present form.

Author Response

Thank you for your detailed and considerated review.